

# Natural wetland methane emissions simulated by ICON-XPP

Stiig Wilkenskjeld[1], Thomas Kleinen[1], Tobias Stacke[1], and Victor Brovkin[1,2]

[1]Max Plank Institute for Meteorology, Hamburg, Germany
[2]CEN, University of Hamburg, Hamburg, Germany

**Correspondence:** Stiig Wilkenskjeld (stiig.wilkenskjeld@mpimet.mpg.de)

**Abstract.** Methane emissions from natural wetlands account for about 1/3 of global methane emissions, and thus have a significant climatic impact due to methane's high global warming potential. Among the sources of methane, those from natural wetlands have the highest uncertainty, and it is thus of key importance to understand and reduce the uncertainties in the estimates of the wetland emissions to be able to close the global methane budget. Using a coupled land and atmosphere setup, a new implementation of interactive wetlands and wetland methane emissions into the ICON-XPP Earth System Model was used to test the sensitivity of the wetland methane emissions to a number of hydrological and biogeochemical model assumptions of which some are mimicing anthropogenic influences on the Earth System. Averaged over the historical period (1855-2014) the simulated emissions are 166.2 (156.3 - 181.2) Tg(CH$_4$) yr$^{-1}$. For the 2000-2012 period the equivalent numbers 182.3 (154.3 - 205.1) Tg(CH$_4$) yr$^{-1}$ are in good agreement with estimates from other studies. Wetland methane emissions rise by about 12% during the historical period, mainly since 1980, an increase which is due to an enhanced carbon cycle caused by the CO$_2$ fertilization associated with rising atmospheric CO$_2$ concentrations. The modeled emissions are very sensitive to changes and assumptions in the model hydrology, some dependencies only revealed through the interaction with the atmosphere (changes in the moisture recycling patterns). Therefore offline land models are only of limited value to test the influence of changes in model hydrology, which is also influenced by changes in terrestrial vegetation.

## 1 Introduction

Methane is — after CO$_2$ — the second most important greenhouse gas on which humans have a significant direct impact (Zhang et al., 2017; Saunois et al., 2016, 2020, 2024). Since the mid 1980'ies, the atmospheric methane concentration has been rising by about 7 ppb per year (Lan et al., 2025) with an increasing trend since 2005. Due to its much higher global warming potential compared to $CO_2$, methane currently contributes almost 1/3 of the anthropogenic climate change (Forster et al., 2021). In combination with the rather short atmospheric lifetime of methane (about 9 years, Saunois et al. (2024)) this could make methane emission reductions an important contributor to reach the goals of the Paris agreement on a decadal time scale (UNEP, 2021).



Methane is emitted to the atmosphere from a number of marine and terrestrial sources of which some are anthropogenic, some natural and some natural but influenced by human activities. Emissions from natural wetlands are the largest of the latter type, as they contribute about 1/3 of the total emissions (Saunois et al., 2016, 2020, 2024). Despite estimate uncertainties, it is commonly agreed, that at least in the recent years, anthropogenic sources have grown larger than the natural ones (Saunois et al., 2024).

Modeling of wetland methane emissions includes many poorly quantified parameters and processes which need to be either estimated from observations or modeled interactively. Since methane production takes place under anaerobic conditions, it is closely coupled to the hydrology of the soils: The performance of the current generation of hydrology models is — positively stated — moderately good (Krysanova et al., 2020; Melton et al., 2013). With few exceptions (e.g. Veldkamp et al. (2018)) land surface models used in global Earth System Models ignore active human water management. Human activities such as dam construction, drainage, irrigation and changes in land use may however severely impact the hydrology on local to regional scales (de Vrese et al., 2018; Xu et al., 2024; Chen et al., 2025). Such measures have historically drained large wetland areas (Davidson, 2014; Gardner and Finlayson, 2018) mainly since the year 1900.

In the high latitudes the presence of (thawing) permafrost further adds uncertainties in wetland estimates (Hagemann et al., 2016; Andresen et al., 2020). The state of the permafrost highly impacts the hydrology of the soils (de Vrese et al., 2023, 2024). Due to small evapotranspiration and the sub-surface runoff partly being blocked by frozen soils, a significant part of global wetlands is located in permafrost areas. Many permafrost-related processes take place at scales far smaller than the resolution of global models (e.g. Rehder et al. (2023)) and depend on subsurface developments, which are presently poorly understood and only possible to observe on local scale. Upscaling these processes to pan-Arctic or even to typical global model grid point scale to obtain robust estimates of soil moisture and wetland area is even more challenging.

Observational estimates of wetland area comes with their own issues and challenges. Zhang et al. (2017) reported a range in the global, yearly mean wetland area from five commonly used different observational products from 5.3 to 10.2 $Mm^2$ (1 $Mm^2 = 10^{12}m^2 = 10^6 km^2$). Also, large disagreement on the global patterns and annual cycle was reported due to differences in methods, observed periods and original purpose of the data set.

Soil methane production requires available soil organic matter (SOM), anaerobic conditions and sufficiently high temperatures, all parameters which are estimated by various model components, all of which comes with their own uncertainties. Over decades, atmospheric and land models have gained good skills in simulating temperatures and carbon content, whereas the oxic state of the soils have until recently only received little attention.

Here we present the results of a new implementation of wetland methane emissions based on interactively calculated wetlands in the ICON-XPP Earth System Model and the sensitivity to different model assumptions about hydrological management and biogeochemical influences. This will hopefully contribute to a better understanding of the controls on the modeled natural methane emission, and thus aims to identify and understand uncertainties in the current estimates of wetland methane emissions.



## 2 Model and development

This study is utilizing the ICON Earth System Model, using the ICON-XPP atmosphere (Müller et al., 2025) and the land model JSBACH4 (Schneck et al., 2022). To enable the model to estimate wetland areas and methane emissions, a number of 60 extensions and modifications to the model code has been implemented. These are described in the following subsections.

### 2.1 Wetlands

Wetland area fraction was estimated using a diagnostic TOPMODEL approach (Beven and Kirkby, 1979), where the compound topographic index (CTI), which describes the relation between the (hydrological) upstream area and the local slope for any point, is used to relate the grid mean water table to the water table distribution on the sub-grid-scale topography. The wetland 65 fraction is then assumed to be the fraction of the grid cell where the water table is at or above the soil surface. The CTI index was obtained as the maximum value of the data set presented in Marthews et al. (2015) and 5. The calculation of the wetland area does not directly feed back to the soil hydrology. It is assumed that the part of the soils, which are diagnosed as inundated using this approach, are anaerobic. Thereby the wetland extend has an influence on the carbon cycle which again influences the hydrology (see Sec. 2.2).

We extended the scheme presented in detail in Kleinen et al. (2020) with two improvements:

– The simple piecewise linear estimate of the ice fraction of soil water used in Kleinen et al. (2020) has been replaced by calculations internally in JSBACH's soil hydrology (Ekici et al., 2014; Hagemann and Stacke, 2015). Thus the freezing and melting of soil water now also influences both the hydraulic conductivity, the energy calculations of the soil and the soil carbon handling.

– In flat areas with shallow soils (e.g. the Sahara), TOPMODEL tends to produce spurious wetlands as a consequence of individual precipitation events, since the shallow soils are rapidly "filled up". To exclude such artificial wetlands while preserving TOPMODEL's capability to react to changing climatic conditions, an additional dynamic condition to exclude wetlands from very dry areas was introduced: We require that

$$\frac{\overline{PR}}{\overline{AET}} > 0.3 \tag{1}$$

were $PR$ is precipitation, $AET$ is the equilibrium evapotranspiration, defined by Eq. (2) and an overline indicates a temporal mean over a "long" period (in this study defined as a running average with an e-folding time of 1 year). This criteria excludes the driest areas while still letting the wetlands develop freely in the rest of the world.

The equilibrium evaporation $AET$ is, following Prentice et al. (1993), defined as:

$$AET = \frac{1}{\lambda} \frac{s}{s + \gamma} NetRad \tag{2}$$

with the rate of change of the saturated vapor pressure: $s = 2.503 \cdot 10^6 \frac{e^{17.269 \frac{t}{t_{ref}}}}{t_{ref}^2}$, the psychrometer constant: $\gamma = 65.05 + 0.064t$, the latent heat of evaporation: $\lambda = 2.495 \cdot 10^6 - 2380t$, $t_{ref} = t + 237.3$, $t$ is the 2m air temperature in $^oC$ and the $NetRad$ is the net radiative balance in $Wm^{-2}$.





## 2.2 Wetland methane emissions

The wetland methane production model implementation is adopted from Kleinen et al. (2020), which is based on the approach
by Riley et al. (2011). The basic assumption is, that the decomposition of SOM in anaerobic soils both produces $CO_2$ and
methane. The anaerobic decomposition pathway is much slower than the aerobic and thus the SOM is changed differently
depending on the oxic state of the soil. Since the soil properties depend on the SOM content, this constitute an indirect
feedback to the hydrological cycle. Following Kleinen et al. (2020, 2021); Wania et al. (2010), we assume a time and space
independent reduction of SOM decomposition rate, so that wetland SOM is decomposed with 35% of the aerobic efficiency.
The fraction of total decomposed SOM converted to methane (the rest is assumed to become $CO_2$) is determined by a Q10
formulation, with 24% methane fraction at 295K and a Q10 value of 1.2.

The produced methane is transported vertically through the soil column via diffusion, ebullition and plant aerenchyma. On
it's way it may be oxidized partly or completely using oxygen which is diffused downward through the soil column. The
oxidation of methane is also parameterized using a Q10 formulation with a Q10 value of 1.9.

## 2.3 Anthropogenic drainage of croplands

To maximize agricultural gains, it has long been practiced to regulate the soil hydrology of crop areas (e.g. Valipour et al.
(2020)). This involved draining excess water away from crop areas. This practice has been build into the model by prohibiting
wetlands on crops. In the current implementation, the drainage affects only the wetland area and not the basic soil hydrology
below crops. To include drained crops furthermore has the advantage that methane emissions from rice fields (Saunois et al.,
2020) are unequivocally attributed as an anthropogenic source and thus falls out of the scope of this study.

## 2.4 Surface water retention (SWR)

The standard JSBACH model uses the ARNO-scheme (Hagemann and Gates, 2003; Hagemann and Stacke, 2015) to separate
rain and melting snow into surface runoff and infiltration into the soil based on the slope distribution of the grid cell. This
scheme ignores that water might pile up in small scale topographic depressions forming ponds or puddles, from where the
water is available for evapotranspiration, evaporation or later infiltration into the soils.Thus it may suppress important water
and energy feedbacks between surface and atmosphere. It is assumed, that in addition to the micro-topographical effects, this
water retention takes place were local soil surface hydraulic conductivity is low. Therefore the conductivity of these areas is
assumed to be equivalent to that of clay, which is the natural soil type with the lowest conductivity. To account for these effects,
we implemented the SWR scheme described in detail in de Vrese et al. (2021); Stacke and Hagemann (2012).

It is assumed that the local SWR is a temporary, though possibly repeated, phenomenon compared to the wetlands arising
due to saturated soils (Sec. 2.1). The area inundated by SWR is therefore not counted as wetlands. Due to the lower infiltration
in areas with SWR, the soil is comparatively dry, and thus we assume that these soils stay oxic and do thus not contribute to
the wetland methane production.



## 3 Setup, experiments and boundary conditions

The experiments were done with ICON-XPP (Müller et al., 2025) with the extensions discussed in Sec. 2. ICON uses a trian-
gular grid and was in this study run at the R2B4 resolution, which has a cell size of about 25,000 $km^2$, roughly corresponding
to a resolution of 160 km in a grid with square cells. The atmospheric model used 90 vertical layers, where the upper layer
has a pressure of about 1 Pa, and the land model contained 5 soil layers with in creasing thickness down to a total soil depth
of at most $\approx$ 9.5 m, though generally the soil is much shallower. The model was run using the concentration driven AMIP
configuration — that is: land coupled to the atmosphere, while ocean sea surface temperature, sea ice and greenhouse gas con-
centrations are prescribed (sea surface temperature and ice using data described in Taylor et al. (2000)). Due to the prescription
of the greenhouse gases — mainly $CO_2$ and methane — there is no direct feedback from the terrestrial carbon cycle on to the
atmosphere. Thus this study does not deliver any estimates of atmospheric concentration or lifetime of the methane.

   The model was spun up from an initial state without carbon for 160 years, then the slow humus soil pools of the YASSO
soil carbon model (Goll et al., 2015; Tuomi et al., 2009) of JSBACH were equilibrated, and the model was spun up for another
50 years. The spinup was done using the vegetation distribution from the LUH2 data set (Hurtt et al., 2020) and greenhouse
gas concentrations (GHGs) for the year 1855. During the spinup the SWR scheme was switched on and crop areas drained as
described above.

   From this spinup, five experiments (Tab. 1) were started, all running from 1855 to 2014, to get a best estimate of the
natural wetland methane emissions and to explore the influence of different model assumptions. The "Base" experiment is a
direct continuation of the spinup, using the SWR scheme and crop drainage, but it switches to transient GHGs and annually
prescribed land use maps derived from the LUH2 data. Other experiments switch off the SWR scheme ("DrySurf"), the
drainage ("WetCrops"), the land cover changes ("ConstVeg") or the transient $CO_2$ concentration seen by the vegetation model,
while preserving the transient GHGs used in the atmospheric model ("ConstCO2") respectively.

The experiments are stopped in 2014 due to lack of newer boundary data for the atmospheric model. Parts of the analysis
therefore concentrate on the period 2000-2012 which was the main focus of Saunois et al. (2016, 2017).

   In this study only the data from the terrestrial areas are analyzed, and thus the term "global" hereafter only refers to the
terrestrial surface. Furthermore, often the results are discussed separately for a number of regions (Fig. S1). The most often
used regions are the northern extratropics (NXT), defined as the area north of $30^oN$, and the tropics, which (unless otherwise
explicitly stated) — following Saunois et al. (2017) — covers the rest of the world. The latter definition is justified since the
southern extratropics only contribute an insignificant fraction of the global wetland ares and wetland methane emissions.

   The LUH2 data set prescribes $5.8 Mm^2$ of crops in 1855 ($3.6 Mm^2$ located in the NXT area). Globally the crop area
increase almost linearly to $15.1 Mm^2$ in 2014. However, the NXT contribution stabilizes at about $7.9 Mm^2$ around 1960 and
decreases after around 1990 to about $7.1 Mm^2$ in 2014. The increase in crop area is wide-spread in non-mountainous areas of
all continents, the exceptions being the north eastern US and Western Europe, where crop areas decrease in the last part of the
experiment period. The crop distribution and its changes are important for a proper interpretation of the results from several of
the experiments.



## 4    Results

### 4.1    "Base" experiment including SWR

The average July–August (JJA) wetland distribution (Fig. 1a) shows the main wetland areas in the tropical rain forests of South America and Africa as well as in the Hudson Bay area and central northern Siberia. Smaller but still important areas are found in India, eastern North America and larger parts of northern Siberia. In addition, some wetlands are found in central western Europe, south-eastern US, Indonesia and eastern China. Roughly, the majority of the northern hemisphere summer wetland area is concentrated in two latitudinal bands: The boreal zone ($55^oN$ - $70^o$) contributing with 34% of the global area and the in

tropics ($20^oS$ - $15^oN$) with 38%. The global JJA wetland area is $\approx 5.2 Mm^2$ with a slight decrease over time (2% per century). This trend over the experiment period, however, hides that the wetlands are only diminishing in the first part of the experiment, while they are growing again after around 1980 at a rate of 7% per century. This increase is mainly driven by changes in the NXT area, though also in the tropics the trend turns from negative to positive over the experiment. Actually the wetland area in NXT region is slightly higher in the latter part of the experiment than in the experiment long average, and is thus opposing the

global decreasing trend (Fig. 4a), so that the NXT fraction of the global wetland areas increase from $< 30\%$ in the 1940'ies to 33% in the last decade of the experiment (2005-2014). In the start of the experiment, this fraction was about 31.6%.

The geographical trend distribution (Fig. 1b), shows areas of both increasing and decreasing wetlands: Decreasing wetlands are found in the east-central US, eastern Europe, southern Brazil and India/Bangladesh, while increasing wetlands are found in eastern US and western Europe. In general there is a close relation between the development of cropland (and thus drainage,

Fig. S3) fraction and the development of the wetlands.

The average simulated wetland area on non-snow-covered areas in "Base" is $\approx 3.8 Mm^2$ with a pronounced annual cycle (Fig. 2), ranging from $\approx 2.8 Mm^2$ to $\approx 5.4 Mm^2$. The year to year variation on the other hand is small ($1\sigma = 0.06 Mm^2$). The annual cycle stems only from the NXT areas, with maximum in NH summer, June to September, where the northern extratropics are snow free. Though the NXT wetland area in the non-NH-summer increase by up to almost $1 Mm^2$ (November)

if no snow masking is applied, the shape of the yearly cycle remains. In the yearly average, snow is covering $\approx 0.5 Mm^2$ of wetlands. The lack of an annual cycle in the tropics is to some degree a result of a cancellation of the annual cycles on the difference continents being out of phase (Fig. S2). The rather weak South American wetland annual cycle peaks in April–July and has its minimum in October–January, quite opposite to somewhat more pronounced annual cycle in Africa, which is the other main tropical wetland region. South East Asia and tropical North America contributes only with very small wetland areas,

leaving their influence on the annual cycle diminutive.

Global wetland methane emissions averaged over the entire experiment period amounts to 166.2 Tg($CH_4$) yr$^{-1}$, of which 35.2 Tg($CH_4$) yr$^{-1}$ or about 21% are emitted in the NXT area. The rest stems almost exclusively from the tropical rain forest areas in South America, Africa and Indonesia, together with India (Tab. 3 and Fig. 3a). The relation between the boreal and tropical wetland latitudinal bands is thus quite different with respect to methane emissions: The boreal band contributes only

8% while the tropical band emits 67% of global wetland methane. Nzotungicimpaye et al. (2021) report a similar distribution of wetland methane emissions between the boreal and tropical regions using a more microbiologically detailed methane model.





For the period 2000-2012 the global average wetland methane emission is 182.7 Tg(CH$_4$) yr$^{-1}$ in close agreement with the multi-model mean presented in Saunois et al. (2016). However, our emissions are more scattered, most pronounced in the Eurasian boreal areas. Higher methane emissions are found in western Europe and southeastern US and lower emissions are found in South East Asia, eastern Africa and the Hudson Bay area. These differences in patterns are quite similar to the multi-model mean of wetland fractions presented in Hardouin et al. (2024).

The emissions are increasing over time with a trend over the entire experiment of about 20 Tg(CH$_4$) yr$^{-1}$ (8% per century). After around 1980, the trend is about 4 times as large compared to the entire experiment (Tab. 3, Fig. 4d). The global increase is mainly driven by increases in the large production areas in South America and Africa as well as Western Europe (Fig. 3b). Decreasing emissions are found in the Central US.

The standard deviation (obtained from the time series detrended by subtracting the linear regressions in the time periods 1855-1980 and 1981-2014 separately. The result show very little effect on the exact point of partition of the regression periods and is virtually identical between the two periods.) of the yearly average emissions is $\approx 3.4$Tg(CH$_4$) yr$^{-1}$.

The emission fraction (the fraction of produced methane that is not oxidized in the soil and thus reaches the atmosphere) is 44.5% and very constant over the experiment period. The emission fraction is geographically very unevenly distributed in complex patterns, where the fraction range from near 0 to almost 80%. In general, the major methane emitting areas have moderate emission fractions of 30-40%, while hardly any methane escapes from dry areas.

### 4.2 No surface water retention ("DrySurf")

Switching off SWR lowers the average wetland area by about $0.1Mm^2$ (Fig. 4a) with the entire reduction taking place in the NXT area, mainly in the period July – October, where the reduction is almost twice the annual mean. Geographically more specific, the reduction is confined to a narrowing stripe reaching from the northeastern Europe turning into central Siberia and the Mackenzie River basin (Fig. 5a).

The areas of reduced wetlands are within huge regions where the precipitation is much lower in "'DrySurf" than in "Base" (Fig. 5b). In parts of central Siberia the precipitation reduction is more than 30%. Most of the areas with large and consistent precipitation differences are in the areas of the NH jet stream. This indicates a huge shift in the magnitude of moisture recycling through the atmosphere, which is supported by a large decrease of evapotranspiration in "DrySurf" compared to "Base" (not shown).

In "DrySurf" the NXT wetland area decreases by about 7% per century and thus 5 times faster than in "Base" (Fig. 4c), whereas the tropical wetlands changes a bit less in "DrySurf" than in "Base". The geographical patterns of changing trends between "Base" and "DrySurf" is coinciding almost exactly with those of the absolute wetland area difference. The average wetland methane emissions (Fig. 4c) from "DrySurf" behave very similar to the wetland area: Slightly lower ($\approx 2.2$ Tg(CH$_4$) yr$^{-1}$) less from the NXT than in "Base" is partly compensated by an even smaller increase ($\approx 0.8$Tg(CH$_4$) yr$^{-1}$) from the Tropics.

In the NXT the decreasing wetland area trend dominates over the generally increasing trend in methane emissions when looking at the entire experiment: A trend of -3% per century opposes the tropical trend of +8% per century. The result is that





the global trend is slightly below that of "Base". Since 1980 this picture is reversed: NXT emissions increase by 42% per century compared to 30% in the tropics.

### 4.3 Neglecting $CO_2$ fertilization ("ConstCO2")

In AMIP runs with prescribed $CO_2$, there is no direct feedback from the terrestrial carbon cycle on to the atmosphere. Thus

it could be expected that the wetlands in "ConstCO2" would be identical to those in "Base". This seems to be confirmed by the average wetland area (Fig. 4a), where the two experiments are indeed very close. The trends over the 1855-2014 period, however, are somewhat different, with slightly smaller trend (less negative) all over the world. In the recent period, the NXT region and the tropics behave very differently: In the tropics the wetland area is stagnating with hardly any trend, while the positive NXT trend in "ConstCO2" exceeds that of "Base". These changes arise as a feedback through changed water use

efficiency of the plants at different $CO_2$ levels, changing the transpiration. Thus both water and energy fluxes to the atmosphere are changed, which then triggers different atmospheric responses. Though the global numbers of evapotranspiration are very similar in the two experiments, both the differences in the evapotranspiration itself and its trend show complex regional patterns (Fig. S5).

Keeping the $CO_2$ concentration constant at pre-industrial levels for the carbon model has a huge impact on the trend of the

methane emissions (Fig. 4d). The increasing methane emissions in "Base" are turned into a decrease in the "ConstCO2" and the trend is thus in the other directions compared to all other conducted experiments. Only the NXT area has a positive trend in the latter part of the experiment, and this is still smaller than in all other experiments except "ConstVeg" (discussed below). The global carbon litter is — opposite to any other experiment — decreasing after around the year 1900 (Fig. S6). There is still a slight increase in the decomposition of the soil carbon, but only about 5% of that in the other experiments. That less

carbon respires more is due to a warmer climate increasing the respiration rate. The reversed trend in the methane emissions in "ConstCO2" reveals, that the trend found in the other experiments is soley due to an enhanced carbon cycle due to $CO_2$ fertilization. Different effects on the wetland methane emissions from rising temperatures seems to be canceling each other out, at least on the large scale.

### 4.4 Increasing potential wetland area: "WetCrops" and "ConstVeg"

The effect on wetland area of switching off the drainage from crop areas and keeping the vegetation distribution constant at the level of the early industrialization is conceptually the same: The area on which wetlands can be created is increased. In case that the drainage is switched off, all crop areas are made available for wetlands, and in the case of constant vegetation distribution, the crops areas are kept smaller as discussed in Sec. 3. This theoretical behavior is confirmed by Fig. 4, where average "WetCrops" wetlands exceeds those of "Base" by $\approx 0.6 Mm^2$ while "ConstVeg" is adding $\approx 0.2 Mm^2$ to the "'Base"

wetland area. This increase is essentially equally distributed between NXT and tropics. These two experiments are the only ones having a positive trend in the global wetland area throughout the experiments, somewhat lower in "WetCrops" than in "ConstVeg" and mostly from the tropical area. In the 1980–2014 period, the trends in "WetCrops" are much larger than those of "ConstVeg" in all regions. In the NXT regions, the wetlands are even decreasing in "ConstVeg" since they do not profit from





the abandoning of croplands in Western Europe and the USA. In general, the major difference to "Base" is that "WetCrops"
and "ConstVeg" do not participate in the downward wetland area trend, where crops are increasing over the experiment period.

The average methane emissions are also increased compared to "Base". However the emissions from "ConstVeg" (176.2
$Tg(CH_4)\,yr^{-1}$) are closer to "WetCrops" (181.2 $Tg(CH_4)\,yr^{-1}$) than to those of "Base" (166.2 $Tg(CH_4)\,yr^{-1}$) . The explanation
can be found in the methane emission trends, where "ConstVeg" is having the largest trend of all conducted experiments. The
reason for the higher trend is that "WetCrops" only changes the wetland area, while no direct changes to the carbon cycle
compared to "Base" are done. "ConstVeg" changes the vegetation distribution in a way leaving more space for the most
productive vegetation type: The forests. Therefore more carbon is available for methane production in "ConstVeg" compared
to "WetCrops".

## 5    Discussion

### 5.1    Impact of including surface water retention

The large differences between the "Base" and "DrySurf" highlights the sensitivity of the estimates of wetland extend and thus
wetland methane emissions to changes in the soil hydrology. Including SWR regionally dramatically increases the evapotran-
spiration and thus atmospheric water recycling. This also has major effects on the fluxes of latent and sensible heat and thus on
regional temperature. These feedbacks emphasize the importance of coupled (in this case land-atmosphere) simulations, since
such additional water recycling (or lack of it) is not accounted for in offline, land only, setups. Thus offline sensitivity studies
to changes in e.g. soil hydrology characterizations may come to wrong conclusions: In equivalent offline experiments (not
shown) forced with the GSWP3 data set (Dirmeyer et al., 2006), the major effect of enabling SWR was to decrease the water
infiltration into the soil, which again led to a slightly reduced wetland area and thus lower wetland methane emissions. Our
assumption that the soil below water retention areas is as impermeable as clay likely is an exaggeration, so that the differences
between "Base" and "DrySurf" may be an extreme case.
Compared to the TOPMODEL wetlands, the SWR scheme is another way of allowing super-surface water to form which can
be interpreted as wetlands. Both SWR and TOPMODEL-style wetlands arise in areas with high water availability and since
both approaches are based on statistical methods, it is impossible to completely separate the two ways of obtaining surface
water bodies. The correlation between the SWR and wetland areas in the entire "Base" experiment is $\approx 0.3$ without trend.
The assumption that soils below SWR areas are oxic may thus be too simplistic. Since SWR almost exclusively influences
North America north-east of the Mackenzie River, north eastern Europe and north western Siberia (Fig. S7), a (partial) lifting
of this assumption would enhance the Arctic and Sub-Arctic wetland methane emissions compared to the tropical ones. By
assuming that there is no overlap between the SWR and TOPMODEL wetland areas and that annual methane emissions are
linearly dependent on annual mean area with wet surface, a rough estimate of the effect of the assumption, that the SWR areas
do not contribute to methane production, can be obtained. In this case global wetland methane emissions would increase by 42
$Tg(CH_4)yr^{-1}$ or $\approx 25\%$. This includes additional 27 $Tg(CH_4)yr^{-1}$ of emissions from the NXT area, increasing the NXT





fraction of the global emissions from 21% to 30%. Since the assumptions for this estimate are rather unrealistic, these numbers are however regarded as the upper extreme of the consequences of excluding the SWR areas from the methane production.

## 5.2 AMIP setup versus offline land simulations

Using an AMIP setup introduces an atmospheric variability which is different from the observed one. Therefore comparisons
of model results to observations and observation based model results are only possible in a statistical sense over longer periods. Additionally the atmospheric model may in AMIP setups introduce less well known biases compared to observational data sets. For this study, the most important known atmospheric bias is an underestimation of the tropical precipitation by about 29% compared to GSWP3 (Tab. 2). The tropical bias is unevenly distributed between the continents, and is by far strongest over Indonesia and South East Asia, which explains our comparatively low wetland areas and thus methane emissions from
this region. The bias distribution agrees with findings of Müller et al. (2025) and seems thus to be a general property of the ICON-XPP atmospheric model. Common wetland products based on observations (GIEMS-2 (Prigent et al., 2020), WAD2M (Zhang et al., 2021)) find a significant seasonal cycle in the tropical wetland area more or less in phase with the northern boreal wetland areas. That in this study essentially no seasonal cycle is found can be attributed to the severe underestimation of south Asian and Indonesian rainfall and thus diminished wetlands in these regions. Therefore the weak cycle in the Amazonian
region can out-weight the out-of-phase cycles in Asia and Africa, producing a net-zero cycle for the tropics.

## 5.3 Anthropogenic land use changes including drainage of croplands

The use of land use maps compared to land use transitions (which are currently not implemented in JSBACH4) is known to underestimate the influence of anthropogenic land use changes on the carbon cycle (Wilkenskjeld et al., 2014; Stocker et al., 2014). Specifically the release of soil carbon is underestimated. Therefore, especially in the tropics, where shifting cultivation is
a common agricultural practice, the carbon pools may be overestimated. This could also lead to an overestimate of the methane emissions from these regions. Since farmers generally try to avoid wet areas (Valipour et al., 2020), we however assume the effect on the methane emissions to be small.

Assuming that all cropland areas in the model are drained is a strong simplification, especially due to the large-scale dynamics and development of agricultural practices. Former crop areas may be abandoned and become either pastures or completely
be given back to the nature. This, of course, would not mean that drainage systems are removed, and thus the area drained is likely underestimated by the approach of this study. Also, managed forests, which in JSBACH are regarded as natural vegetation, pastures, and areas of urban expansion may be drained to enhance tree growth, making the land more suitable as grassland or to make the ground supportive for technospheric constructions (Fluet-Chouinard et al., 2023). Again these effects will lead to an underestimate of drained area. On the other hand, not all croplands are drained, either because they are dry enough by nature
or because of limited local resources to establish adequate drainage systems. As an example, Valipour et al. (2020) states that in present day India, only about 10% of croplands are drained. This will lead to an overestimate of our drained area, but likely not change the simulated wetlands much, since the extra drained area will anyway be too dry to regularly become wetlands. Though Valipour et al. (2020) reports a history of several thousand years of anthropogenic drainage, Fluet-Chouinard et al.





(2023) argue that main wetland area loss due to drainage takes place after 1900 and is still ongoing at a more or less constant
rate. Thus our assumption of drained crops may be most representative for the latter part of the simulation period. With the exception the NXT region in the "ConstVeg" experiment wetland areas are increasing since 1980 in all our experiments for both NXT and tropics. One reason may be different developments in the precipitation and precipitation patterns. GSWP3 and the ICON-XPP atmosphere agree that in the 1980–2014 period, the global (land) precipitation is on average enhanced ($\approx 0.48, 0.43$ and $0.32 mmyr^{-2}$ for GSWP3, "Base" and "DrySurf" respectively). However, the global patterns of increasing precipitation
in ICON-XPP correlates better with our simulated wetlands (somewhat better in "BestGuess" than in "DrySurf"), which may thus be enlarged. Nevertheless, it is unlikely that this better pattern correlation alone can explain the trends in wetland area going in different directions in our simulations and observations. It is more likely that we, despite the drainage assumption, largely underestimate the influence of human water management (dams, reservoirs, drainage — also of non-cropland). In total we expect that our simulated drained area to be underestimated and the wetland areas therefore overestimated in regions with
intense and long-lasting anthropogenic influence.

### 5.4 Wetland area estimates

Our simulated wetland is approximately in line with those presented in Prigent et al. (2007) when accounting for their data set including also anthropogenic wetlands, mainly rice fields (Kleinen et al., 2020). The estimates from Zhang et al. (2021); Prigent et al. (2020); Lehner and Döll (2004) are however much larger. The new version 2.0 of the WAD2M wetland data
set (Zhang et al., 2021) reduced their wetland extent estimate by 0.5 Mm$^2$ compared to the previous version (Saunois et al., 2024). One reason for our smaller wetland area might be, that the available observational data sets still do not distinguish small lakes from wetlands, whereas the TOPMODEL approach only counts (super-)saturated soils as wetland. Another reason is the ICON-XPP precipitation bias over the tropics — specifically South East Asia, leading to too low an estimate of wetland extent in our experiments. Also JSBACH is not equipped with a detailed river flow model capable of simulating temporary flooding
of river flood plains as a consequence of heavy precipitation or snow melt events. Since such wetlands tend to be short-term phenomena, it is questionable if the soils have time to become anaerobic and thus they would likely not contribute much to the wetland methane emissions.

Excluding short term wetlands from dry areas by applying the dynamic application of the criterion in Eq. (1) instead of a static mask allows the model to catch the influence of major climatic shifts. On the other hand it may also cause spurious trends
in wetland area and thus in wetland methane emissions by changes in the wetland allowance mask, either temporary or as part of a long term trend. By selecting the required ratio as low as 0.3 (Eq. (1)), the bounds of where wetlands are allowed are however far away from where major wetlands are predicted. Thus the wetland methane emissions and their trends are likely not influenced by the details of this masking.

### 5.5 Wetland methane emissions

In the "ConstCO2" experiment the trend in wetland methane emissions is much smaller than in "BestGuess". Over the entire experiment period it is even negative, following the downward trend in wetland area. This suggest that the main driver of



the trend found in all other experiments is an accelerated carbon cycle stemming from $CO_2$ fertilization. This contradicts the findings of both McNorton et al. (2016) and the WETCHIMP model ensemble (Melton et al., 2013), who attributes their trend to temperature driven increase of wetland area and microbial methane production rates.

For the period 2000–2012 the trend in global wetland methane emissions is about 0.4 $Tg(CH_4)yr^{-2}$ of which 75% stems from the tropics. This is higher than the numbers presented in Saunois et al. (2017) who reports a multi-model mean of 0.2 $Tg(CH_4)yr^{-2}$ and about 10% of a trend of 2.2 $Tg(CH_4)yr^{-2}$ from top-down inversions attributed to non-anthropogenic sources. Since the productivity in JSBACH is known to be comparatively sensitive to changes in $CO_2$ atmospheric concentration, which is the driver of our trend in methane emissions, it seems logical that our predicted wetland emission trend is at the
high end.

The basic TOPMODEL assumption defining wetlands as areas with super-saturated soils (i.e. water level is at least reaching the surface) and using the wetland fraction as a measure of the soil oxygen content are simplifications, which can only serve as a first guess. For instance soils where the root zone of macrophytes is saturated may be anoxic. Thus, though the soil column as a whole is not saturated, such areas could be a source of methane mainly since produced methane may bypass the upper (oxic)
soil layers through aerenchyma. On the other hand, soils takes time to become anoxic after being inundated. Depending on the state of the soil at inundation, this may take hours to weeks Patel et al. (2024), and the anaerobic area could thus potentially be overestimated by assuming all wetland soils to be anoxic. Also wetland areas may have very different and highly heterogeneous methane production and emission rates as shown for Arctic ponds by Rehder et al. (2023).

The tropical precipitation bias reducing our tropical wetland area of course tends to decrease the importance of the tropical
wetland methane emissions. There are however also effects which draws in the other direction: In this study, we accounted only for the conversion of contemporary organic matter to methane, while in reality an additional source may be added from the huge amount of ancient organic carbon stored in the thawing permafrost (Hugelius et al., 2014) as well as from degrading peatlands (Hugelius et al., 2020), since we include no peatland model such as HIMMILI (Raivonen et al., 2017). Ekici et al. (2019) found that incorporating micro-scale ground subsidence following permafrost thaw will increase the wetland fraction
and thus methane emissions. Both these effects would enhance the importance of the Arctic regions in the global methane budget. Even though our wetlands are in general smaller, the modeled NXT fraction agrees with the more detailed wetland methane model of Nzotungicimpaye et al. (2021).

## 6    Conclusions

Sub-models for interactive wetland dynamics and wetland methane emissions have been build into the ICON-XPP Earth Sys-
tem Model. The resulting wetland distributions, when accounting for precipitation biases from the atmospheric model, align very well with observations. Additionally, the estimates of wetland methane emissions agree well with published values. Sensitivity studies were done to quantify effects of certain model assumptions and anthropogenic influences on methane emissions from natural wetlands.



Agricultural development encompassing both the changes in farmed area and cultivation practices, is the dominant anthro-
pogenic factor controlling the natural wetland methane emissions by controlling the potential wetland area. Until around 1980
this result in a global decrease in wetland area, counteracting the general trend of increasing wetland methane emissions. In
contrast to the simulations here presented, observations suggest that global wetland area is still decreasing as a consequence
of human actions, indicating that anthropogenic influences on the global soil hydrology are underestimated in the current
generation of Earth System Models.

The second most significant humanly-influenced factor affecting wetland area and their methane emissions is the increase
in atmospheric $CO_2$ concentrations. This rise accelerates the carbon cycle through $CO_2$ fertiliazation. This, however, does not
significantly impact the wetland area. The temperature effect of rising atmospheric $CO_2$ concentration does not substantially
affect wetland area and methane emissions.

Effects of changing how the the surface hydrology is modelled, including the implementation of surface water retention, has
the potential to significantly alter land-atmosphere water and energy fluxes on the regional scale and thus needs to be evaluated
using coupled setups.

*Code and data availability.* Code, scripts and data used for this manuscript as available in the Edmond repository, Wilkenskjeld (2025).

*Author contributions.* S. Wilkenskjeld ported the wetland and methane models to the ICON-ESM and developed the model extensions,
designed and conducted the experiments as well as the output analysis and did the main work on writing this paper. T. Kleinen implemented
the original wetland and methane models in MPI-ESM and provided much insights into wetland and methane cycling in the Earth System.
T. Stacke implemented the SWR scheme and contibuted his knowledge on soil hydrology and its interplay with the atmosphere. V. Brovkin
contributed the idea of the project and gave it directions during the process. All authors contributed to the writing process of the paper.

*Competing interests.* The authors declare that they have no competing interests.

*Acknowledgements.* This study was funded by the EU-Horizon 2020 projects ESM2025, Grant-no: 101003536 and Q-Arctic (ERC-grant-
no: 951288) and used resources of the Deutsches Klimarechenzentrum (DKRZ) granted by its Scientific Steering Committee (WLA) under
project ID bm1255.



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





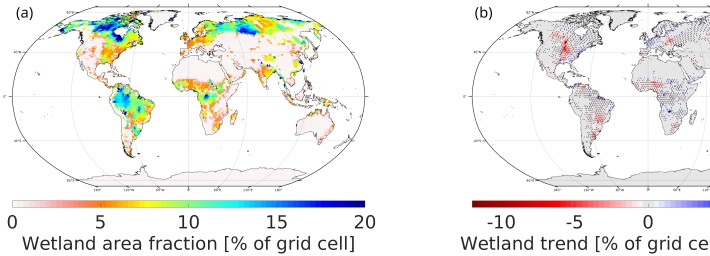

**Figure 1.** "Base" average distribution of (a) wetland area (% of grid cell area) in northern hemisphere summer (JJA) and (b) 1855-2014 wetland area trend of yearly means (% of grid cell per century). Black dots mark cells with trends significant at the 95% level according to Matlab's fitlm function.

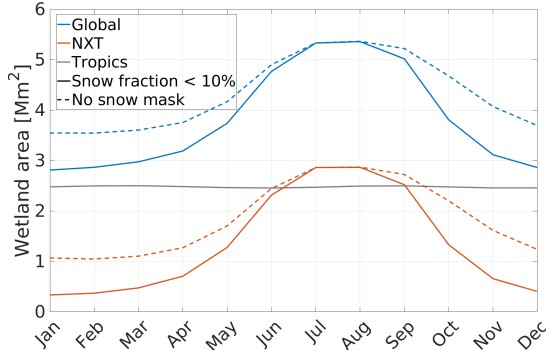

**Figure 2.** Wetland area yearly cycle for the "Base" experiment. Solid lines only include cells with less than 10% snow whereas dashed lines includes all cells. Blue lines are global values, reddish include only the NXT area (north of $30^o$N) and the gray line is the "'tropics" (everything south of $30^o$N). For the tropical area, snow masking does not make any difference.

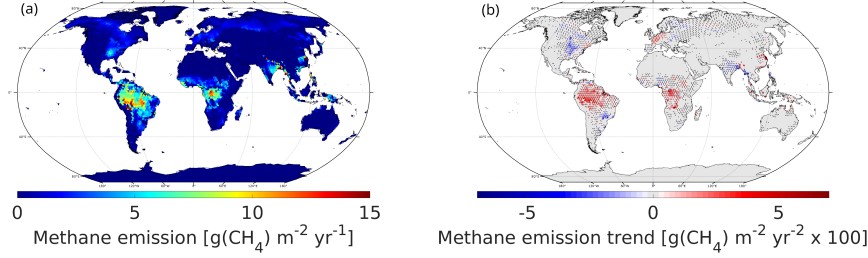

**Figure 3.** Global distribution of (a) wetland methane emissions in the "'Base" experiment, 1855-2014 average and (b) trend of wetland methane emissions in the "Base" experiment. Black dots mark cells with trends significant at the 95% level according to Matlab's fitlm function.





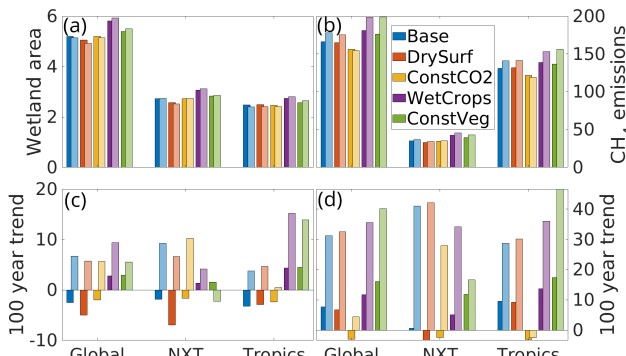

**Figure 4.** Top panels: NH summer (JJA) Wetland area (in $[Mm^2]$) for areas with $\leq 10\%$ snow (a) and annual wetland methane emissions (b, $[Tg(CH_4)\ yr^{-1}]$). Bottom panels: 100 year trend ([% of 1855–2014 average]) of (c) annual wetland area and (d) annual wetland methane emissions. Dark (colors exactly matching the legend) partly covered bars represent the entire experiment period (1855-2014), while brighter overlaid bars represent the 1980-2014 period. Colors indicate the different experiments.

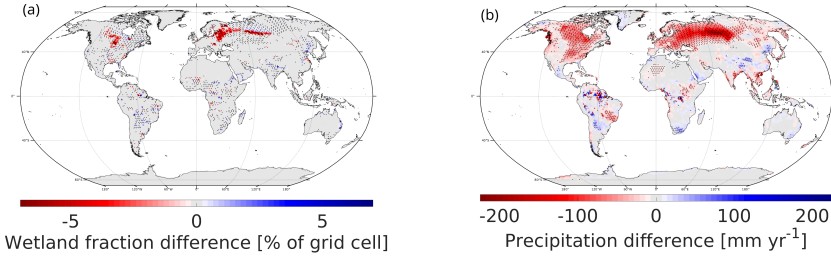

**Figure 5.** Mean difference in wetland area (a) and annual precipitation (b) between "DrySurf" and "Base" . Black dots mark cells significantly different at the 95% level between the experiments according to Matlab's ttest function.

**Table 1.** List of conducted experiments. The $CO_2$ column only applies to the $CO_2$ seen by the terrestrial carbon model. For the radiation calculations, transient $CO_2$ concentration is always applied.

| Name | Type | Ponds | Drainage | Vegetation | CO$_2$ |
|------|------|-------|----------|-----------|-----|
| Base | AMIP | Yes | Yes | Transient | Transient |
| DrySurf | AMIP | No | Yes | Transient | Transient |
| ConstCO2 | AMIP | Yes | Yes | Transient | 1855 |
| WetCrops | AMIP | Yes | No | Transient | Transient |
| ConstVeg | AMIP | Yes | Yes | 1855 | Transient |





**Table 2.** GSWP3 and model precipitation [mm yr$^{-1}$]. Annual means from year 1901 to 2014. "Tropical": latitudes between 30°S and 30°N, "SA": Tropical South America (latitudes 30°S to 12°N), "AF": Africa south of 30°N, "EA": East Asia (latitudes 15°S to 20°N, east of 90°E).

| Dataset | Global | Tropical | NXT | SA | AF | EA |
|---|---|---|---|---|---|---|
| GSWP3 | 776.0 | 1130.4 | 553.1 | 1743.6 | 768.9 | 2376.3 |
| Base | 651.4 | 809.6 | 588.5 | 1316.9 | 633.6 | 964.4 |
| DrySurf | 640.2 | 808.0 | 563.9 | 1316.8 | 634.1 | 944.7 |

**Table 3.** Global and northern extratropics (NXT) wetland methane emissions for the entire experiments, the period 2000-2012 (global only, compare to Saunois et al. (2016)) and wetland methane emissions trends for the entire experiments and the last 30 years of the experiments. Trends are given in Tg(CH$_4$) per year per century.

| Experiment | Global mean 1855-2014 | Global mean 2000-2012 | NXT mean 1855–2014 | Global trend 1855-2014 | Global trend 1980-2014 | NXT trend 1855-2014 | NXT trend 1980-2014 |
|---|---|---|---|---|---|---|---|
| | [Tg(CH$_4$) yr$^{-1}$] | | | [Tg(CH$_4$) yr$^{-2}$ x 100] | | | |
| Base | 166.2 | 182.7 | 35.2 | 12.8 | 55.6 | 0.2 | 15.1 |
| DrySurf | 164.8 | 179.9 | 32.9 | 11.0 | 57.0 | -1.1 | 14.3 |
| ConstCO2 | 156.3 | 154.6 | 34.7 | -4.6 | 6.9 | -0.9 | 9.9 |
| WetCrops | 181.2 | 204.5 | 42.6 | 21.2 | 70.7 | 2.1 | 15.5 |
| ConstVeg | 176.2 | 205.5 | 39.5 | 28.4 | 79.8 | 4.6 | 7.1 |