# Peer review of "Natural wetland methane emissions simulated by ICON-XPP"

_EGUsphere, 2025_

## Author Comment (AC2)

General comments: In this study, the authors incorporated wetlands and wetland methane emissions into the ICON-XPP model to estimate global methane emissions. They tested the sensitivity of the results using a coupled model simulation and identified some dependencies with the atmosphere. Although there have been some studies that have estimated wetland methane emissions using models, these are mostly carried out offline. Therefore, this study is an interesting addition to our current knowledge. However, some aspects need improving before publication. One of the main concerns is that the method requires more detail. For example, the processes of methane production, oxidation and transport, and the model simulation setups. Without these details, the results were difficult to understand. Specific comments follow:

We thank the reviewer for her/his interest and encouragement, as well as the constructive criticism, which we will try to consider as well as possible.

As mentioned in the introduction, human activities significantly modify surface hydrology (line #32). Incorporating all these activities will be a huge challenge, but would it be possible to include wetland drainage? For example, could we include the percentage of the wetland area that has been drained each year since 1900, if this better represents the wetland area?

Thanks very much for this excellent suggestion. This will, however, be very difficult for two reasons: 1) The coupled atmosphere - land surface experiments we have conducted thus far are rather expensive in terms of required computation time, making it very difficult to repeat them. 2) We are lacking data: A few years ago, we looked into the drainage history of peatlands, first on the global scale, and then on the national (Germany) scale. Even on this very limited scale, records were extremely difficult to obtain, and neither comprehensiveness of coverage nor representativeness could be ensured. Thus this is an excellent suggestion, but we will not be able to do this for the revision of our manuscript.

Line #198: Could you add the number from Saunois et al. (2016)?

Yes, of course, although it may become the updated Saunois et al (2025), if there is a number given there.

Figure 4: Could you extend the axis range to make the ends of the bars more visible?

Yes, we will update the Figure.

DrySurf experiment: The SWR scheme changed the surface water dynamics as well as the precipitation patterns. Did including SWR in the model lead to more realistic wetland extent and precipitation patterns?

Overall, including the SWR scheme leads to an improvement in the wetland extent and precipitation patterns. In the revision of our manuscript, we will extend the discussion of the effects of enabling / disabling the SWR scheme.

I find the WetCrop setup a bit confusing: it could be due to the experiment name, the definition of wetlands or the lack of detailed information. Switching off the drainage means that croplands can be

converted to wetlands (with a change in vegetation), increasing both the area of wetlands and their methane emissions. However, what does it mean when you say there are no effects of drainage on the basic soil hydrology below the crops in line #103 and no changes in the carbon cycle compared to the 'Base' in line #259? Are croplands kept wet without any vegetation change from crops to wetland plants? If the croplands are 'wet', why does the wetland area increase together with methane emissions from wetlands?

I fear there is a basic misunderstanding of what we (and other global wetland/methane modellers) mean by the term "wetland", as this is, of necessity, different from what people who are working empirically consider. While a wetland in the ecological sense is an ecosystem that is characterised by wet conditions, as well as specific plant assemblages, this level of detail cannot, at present, be represented in global land surface models. Here, a "wetland" is defined purely hydrologically as an area where the water table is at or above the surface, leading to anaerobic conditions in the soil, and thus methane production. Vegetation, on the other hand, is prescribed from remote sensing in our experiments, and thus does not react to the hydological conditions in the soil.

Thus, in a sense, global modellers tend to use the term "wetland" inappropriately, and in writing our manuscript we made the mistake of not clarifying this sufficiently.

Therefore, if we "switch off" the drainage in the model, the croplands in the model can now be inundated (and thus generate methane, if inundated for long enough for conditions to become anaerobic), but they do not become wetlands in the ecological sense, as vegetation does not change.

In the revision of our manuscript, we will rewrite these sections in order make these distinctions clearer.

The same applies to ConstVeg: is the land cover of 1855 kept constant throughout the simulation despite changes to surface water? If so, how do wetland area and methane emissions change?

See above: In our model experiments, vegetation is prescribed and independent of surface water. If hydrological conditions change, inundation changes (in our manuscript, it would be called a change in wetland area), leading to methane changes.

Section 2.1: What were the temporal and spatial resolutions used to calculate the wetland area fraction? Was the resolution high enough to capture the heterogeneity of permafrost areas such as polygonal tundra or palsa?

Wetland area (better: inundation) is calculated for each model time step, with our time step being 7.5 minutes. We are using a triangular grid in resolution "R2B4", which corresponds to a grid spacing of 160 km between triangle centres. The wetland area, on the other hand is determined from a topographic data set with a resolution of 15 arc-seconds, i.e., roughly 500m. Thus no, the resolution would not be high enough to consider such features.

In the revised manuscript, we will extend Section 2.1 to clarify these matters.

Section 2.2: What are the exact processes of methanogenesis? SOM decomposes to 100% $CO_2$ in aerobic conditions and to 50% $CO_2$ and 50% $CH_4$ in anaerobic conditions with 35% efficiency. What is the proportion between acetoclastic methanogenesis and hydrogenotrophic methanogenesis? Do you assume that acetoclastic methanogenesis is dominant? A more detailed

description of methanogenesis could be provided.

Yes, in the laboratory SOM decomposes 50:50 to $CO_2$ and $CH_4$ under anaerobic conditions. However, results in the field apparently are much more variable. In our modelling approach, we do not actually distinguish between acetoclastic and hydrogenotrophic methanogenesis, as is common practice in global methane models. In a review by Xu et al. (2016), 37 out of 40 models reviewed did not make this distinction. This is a known shortcoming of these models, and a potential source of bias.

In the revision of our manuscript, we will extend the description of methane processes, including methanogenesis, in our model.

Section 4.1: To what extent are the wetlands comparable with those in global and regional inventories? If not, what could be causing the uncertainty? Despite the wetland extent potentially being underestimated, global methane emissions are comparable with those in other syntheses. Is methane emission per area then overestimated? A more detailed description of methane production and transport processes would help us to understand this issue.

Yes, wetlands, or rather inundated areas, are comparable to those in global inventories. However, global inventories (estimates from remote sensing) differ considerably, depending on source, remote sensing product analysed and processing technique. Furthermore, the uncertainty in estimates for model parameters is considerable. As a result, a wide range of global methane emissions would be compatible with model parameter uncertainty, and the particular parameter values were chosen to yield results comparable to observations.

In the revision of our manuscript, we will extend the description of the methane production and transport processes to clarify these issues.

Line #269: I would be careful with the comment that offline simulations can lead to the wrong conclusion. Depending on the purpose and how the model works, offline and online simulations can have different strengths and weaknesses. For example, interactions between wetlands and the climate can only be observed through online simulations. However, if the area of wetlands and wetland methane emissions, as well as their interaction with the climate, are not simulated well, there can be higher errors in online simulations than in offline simulations, where the extent of wetlands and the climate are well constrained.

Thanks for this reminder, we will formulate this more carefully and discuss these caveats.

Section 5.4: Could you add the numbers of wetland extent from each reference?

Yes, we will include those numbers.

There are many typos. These are the ones I found, but there may be more. Please read through the manuscript carefully and correct them.

1980'ies to 1980s in line #18
$CO_2$ to $CO_2$ in line #20
wetland extend to wetland extent in line #68

it's to its in line #98

were to where in line #112

in creasing to increasing in line #123

1940'ies to 1940s in line #165

the paragraph needs to be organized better to avoid two sentences in parenthesis in line #196

We correct all of the typos mentioned here, and we will carefully check the entire manuscript for further typos.

References:

Riley, W. J., Subin, Z. M., Lawrence, D. M., Swenson, S. C., Torn, M. S., Meng, L., Mahowald, N. M., and Hess, P.: Barriers to predicting changes in global terrestrial methane fluxes: analyses using CLM4Me, a methane biogeochemistry model integrated in CESM, Biogeosciences, 8, 1925–1953, https://doi.org/10.5194/bg-8-1925-2011, 2011.

Saunois, M., Martinez, A., Poulter, B., Zhang, Z., Raymond, P. A., Regnier, P., Canadell, J. G., Jackson, R. B., Patra, P. K., Bousquet, P., Ciais, P., Dlugokencky, E. J., Lan, X., Allen, G. H., Bastviken, D., Beerling, D. J., Belikov, D. A., Blake, D. R., Castaldi, S., Crippa, M., Deemer, B. R., Dennison, F., Etiope, G., Gedney, N., Höglund-Isaksson, L., Holgerson, M. A., Hopcroft, P. O., Hugelius, G., Ito, A., Jain, A. K., Janardanan, R., Johnson, M. S., Kleinen, T., Krummel, P. B., Lauerwald, R., Li, T., Liu, X., McDonald, K. C., Melton, J. R., Mühle, J., Müller, J., Murguia-Flores, F., Niwa, Y., Noce, S., Pan, S., Parker, R. J., Peng, C., Ramonet, M., Riley, W. J., Rocher-Ros, G., Rosentreter, J. A., Sasakawa, M., Segers, A., Smith, S. J., Stanley, E. H., Thanwerdas, J., Tian, H., Tsuruta, A., Tubiello, F. N., Weber, T. S., van der Werf, G. R., Worthy, D. E. J., Xi, Y., Yoshida, Y., Zhang, W., Zheng, B., Zhu, Q., Zhu, Q., and Zhuang, Q.: Global Methane Budget 2000–2020, Earth Syst. Sci. Data, 17, 1873–1958, https://doi.org/10.5194/essd-17-1873-2025, 2025.

Xu, X., Yuan, F., Hanson, P. J., Wullschleger, S. D., Thornton, P. E., Riley, W. J., Song, X., Graham, D. E., Song, C., and Tian, H.: Reviews and syntheses: Four decades of modeling methane cycling in terrestrial ecosystems, Biogeosciences, 13, 3735–3755, https://doi.org/10.5194/bg-13-3735-2016, 2016.